# Influence of School Type and Class Level on Mean Caries Experience in 12-Year-Olds in Serial Cross-Sectional National Oral Health Survey in Germany—Proposal to Adjust for Selection Bias

**DOI:** 10.3390/ijerph21040467

**Published:** 2024-04-10

**Authors:** Julian Schmoeckel, Goetz Wahl, Ruth M. Santamaría, Roger Basner, Elisabeth Schankath, Christian H. Splieth

**Affiliations:** 1Department of Preventive and Pediatric Dentistry, University of Greifswald, Fleischmannstr. 42, 17475 Greifswald, Germany; ruth.santamaria@uni-greifswald.de (R.M.S.); splieth@uni-greifswald.de (C.H.S.); 2Landesamt für Verbraucherschutz Sachsen-Anhalt (LAV), Große Steinernetischstr. 4, 39104 Magdeburg, Germany

**Keywords:** caries experience, epidemiology, selection bias, adjustment model, oral health, children

## Abstract

The objective of this study is to analyse the effects of attended school type and class level on the reported caries experience (DMFT) obtained in the serial cross-sectional National Oral Health Study in Children in Germany (NOHSC) for the WHO reference group of 12-year-olds. Methods: Caries data from the 2016 NOHSC were adjusted for each federal state on the basis of two additional large-scale datasets for school type and class level. Results: Twelve-year-olds in all grades in Saxony-Anhalt (n = 96,842) exhibited significantly higher DMFT values than 12-year-olds in 6th grade (n = 76,456; +0.10 DMFT; ~14.2%, *p* < 0.001). Adjustments for school type had effects on DMFT on the level of federal states but almost balanced out on the national level (−0.01 DMFT; ~2%). Due to putatively similar structures of the federal states, the national mean DMFT for 12-year-olds in the latest NOHSC (2016; n = 55,002) was adjusted from 0.44 to 0.50 DMFT, correcting for selection bias. Conclusion: Selection bias in this NOHSC leads to an underestimation of caries levels by about 15%. Due to very low caries experience in children in Germany, these precise adjustments (+0.06 DMFT) have only a minor effect on interpretations of the national epidemiologic situation. Consequently, other national caries studies worldwide using the robust marker of DMFT should also adjust for systematic selection bias related to socio-economic background rather than increasing efforts in examination strategy.

## 1. Introduction

A clear decline in caries levels in the permanent dentition in children has been observed in the last decades in Germany [1,2], similarly to other countries [3,4,5]. Caries data from countries worldwide with likely large differences in methodology have been and still are put together to describe the oral health situation [5,6]. No doubts exist with regard to these trends in caries experience, but doubts about the representativeness of caries studies in 12-year-olds still exist, which might affect the comparability of the caries data (DMFT) as a wide range of different methods, including calibration and caries criteria, were used [7]. Despite the generally high effort applied to select a representative sample to achieve a high quality in these epidemiological studies, selection bias still likely occurs in any study. Selection bias describes a systematic difference between participants in the study and non-participants, which affects the generalisability of the outcomes and should be of concern in any epidemiological study [8]. This means that these systematic errors in the research methodology may also have an impact in the case of the German National Oral Health Survey in Children (NOHSC) [9,10].

The consecutive cross-sectional German NOHSC performed from 1994 to 2016 was initiated by the German Association for Dental Prevention in Children (Deutsche Arbeitsgemeinschaft für Jugendzahnpflege e. V.—DAJ) and its 17 regional bodies (LAJs) [9]. It showed a considerable caries decline in the WHO reference group of 12-year-olds [11]. Due to practical reasons, the sampling in 12-year-olds was restricted to 6th grade only, as it was assessed that about 70% of the 6th-graders were 12 years old [9]. For examinations of (almost) all 12-year-olds in Germany, it was determined that all classes from 4th to 8th grade would need to be visited, which would lead to tremendously higher logistical and costly efforts. Still, this convenient examination strategy could create a potential selection bias leading to systematic error in the reported values in caries experience (DMFT), especially if socio-economic parameters are involved, which are known to have a strong impact on oral health not only in Germany [9,12]. While factors such as the influence of different types of schools at a certain age (e.g., 12 years) on the caries scores have been reported in the literature [1,2,11,13], the influence of attending different class levels at the same specific age are not well researched. Moreover, the aspect of selection bias involving socio-economic aspects, such as those depicted by the attended class level and type of school, needs to be investigated to check for the validity of the results in national oral health studies.

We hypothesize that selection bias occurred in the GNOHS 2016 as (a) caries experience in 12-year-olds strongly depends on the attended type of school, and (b) 12-year-olds in 6th grade in Germany have lower DMFT values than in other class levels. Socio-economic aspects likely play a role as the choice of a certain school form—especially in Germany—depends on the social status of the child’s family, and social status is associated with dental health [9]. Children with a low socio-economic status (SES), who generally have poorer dental health and oral health related quality of life [14], more often attend institutions with a lower level of education, whereas children with a higher SES, who generally have better dental health, more often attend institutions with a higher level of education. Regarding the aspect of attended class level, we believe that 12-year-olds in a grade <6 have a higher DMFT on average, because many of them are students who retake a class, with lower SES being again very closely associated with poorer dental health. The effect of SES may outweigh the effect of a slightly younger age (lower DMFT) in this specific case. On the other hand, we believe that 12-year-olds in a grade >6 also have a higher DMFT on average because statistically, they have a higher age per month than 12-year-olds in grade 6 and each additional month of life increases the (biological) caries risk. The age effect here rather outweighs the social effect because social status-related early enrolment/class skipping (young 12-year-olds with high social status who are enrolled early or have skipped a class and have a lower DMFT due to high social status) is less frequent than SES-related repetition of a school year.

Fortunately, in 2016, due to routine, calibrated, annual oral examinations in schoolchildren by the community dental services in the German federal state of Saxony-Anhalt, a full survey of a large fraction of all 12-year-olds was available parallel to the NOHSC sample of 12-year-olds in 6th grade [9], which allowed for a pilot assessment on the described effects. In Saxony-Anhalt, routine dental examinations are repeated by the public health service on a yearly basis alongside the caries preventive visits in schools. This special situation at the level of a federal state offers the unique possibility to assess the potential effect and magnitude of the bias restricting examinations to certain class levels in a time frame of more than ten years, assessed prior to the COVID-19-pandemic. This led to the idea to use pooled data from this region for a stronger underlying data basis. Additional selection bias was to be assessed via available data from the Federal Statistical Office of Germany, reflecting the true distribution of children attending different types of school in Germany [15], which are known to be correlated with significant differences in DMFT values [9].

Therefore, the purpose of this study is to evaluate the effect of attended school type and class level on the reported caries experience (DMFT) obtained in the National Oral Health Study in Children in Germany (NOHSC) for the WHO reference group of 12-year-olds in all federal states of Germany. The proposed adjustment strategy should reveal precise assessments and improve interpretation of the selection bias in large-scale epidemiological national studies on oral health and may give an insight into the necessity of fine-tuning examination strategies in caries epidemiology vs. statistical adjustments.

## 2. Materials and Methods

### 2.1. Design

The German National Oral Health Survey in Children (NOHSC) was designed as a large-scale, consecutive, epidemiological, cross-sectional study on oral health in children in Germany since 1994, initiated by the German Association for Dental Prevention in Children and Adolescents (DAJ) [2,9]. The school examinations were executed on the basis of WHO methods [16], and the national guidelines of the Federal Association of Dentists of the Public Health Service (BZÖG) [17] were followed. The study was approved by the Ethics Committee of the University of Greifswald (Reg.-Nr.: BB48/10a) and conducted in accordance with the principles for medical research involving human subjects described by the Helsinki Declaration (1964) and its later amendments. This human observational study also conformed to the STROBE guidelines (STROBE checklist).

Since 1994, one important obligatory target group for oral examinations has been the 12-year-olds in all different types of schools (WHO reference group). Due to practical reasons within the NOHSC, only the ones attending 6th grade were examined. The aim of the NOHSC was to assess oral health in all German federal states individually, with a high representativeness (regarding sampling on school level) relating age and region and, therefore, also comparability to all German federal states and caries trends for the country in this age group. The oral examinations in the latest NOHSC took place in the schools during the 2015–2016 school year (examination time period: 1 August 2015–31 July 2016) [9]. The inclusion criteria for children to be examined within the NOHSC were as follows: (a) attending 6th grade, and (b) being 12 years of age at the date of routine examination by a public health dentist in the school. More information e.g., sampling methodology, can be read elsewhere [9].

In a pilot analysis, the data on 12-year-olds in 6th grade from Saxony-Anhalt and the other federal states (n = 55,002 Germany in total) examined within the NOHSC were compared to the caries data of all 12-year-olds examined in Saxony-Anhalt regardless of the class level during the same time frame (school year 2015/2016), which were additionally provided by the Landesamt für Verbraucherschutz Sachsen-Anhalt [9]. To confirm the preliminary data from the pilot study, the relationship between the DMFT of 12-year-olds in 6th grade and the DMFT of 12-year-olds in all grades was explored for this study in an even larger dataset (2008–2018; n = 96,842) from Saxony-Anhalt (Table 1).

In addition to the selection bias due to the restriction of examination to only class level 6, data of the actual distribution of schoolchildren in 6th grade in relation to the type of school were used [15]. Due to a great variety of school types in the different federal states, these were grouped together for the sake of generalisability and simplicity to only two major categories: “Gymnasium” (mainly schoolchildren of higher learning performance and more privileged social background) vs. all other schools (Table 2). A comparison of the percentages shows federal states with a comparable rate (within five percentage points) between population and sampling, but also regions with higher differences, which highlights that the convenient examining strategy may have a significant impact on DMFT in certain federal states.

#### Overview on Used Datasets

To summarize, for this study, three different large datasets are used:(a)data of the latest NOHSC from the 2015–2016 school year (n = 55,002) containing caries data of 12 year-olds in 6th grade in all federal states in Germany [9,10];(b)assembled data from the 14 public dental services in Saxony-Anhalt from the annual oral examinations accompanying group prophylaxis in schools from the years 2008 to 2018 for all 12 year-olds irrespective of class level (n = 96,842, Table 1) [18]; and(c)data from the German federal statistical office on the distribution of schoolchildren dentists in Saxony-Anhalt examined 6391 12-year-old 6th-graders, which is about 68% of in 6th grade to the different types of schools in each federal state (Table 2) [15,19].

Dataset (a) is adjusted with the findings from (b) and (c) to assess the magnitude of potential selection bias in NOHSC and its effects on caries data.

### 2.2. Study Sample in Saxony-Anhalt

In Saxony-Anhalt, caries data were extracted from the routine oral health examinations of the public dental service in the 2015–2016 school year for which all 12-year-olds were sought to be examined (obligatory school examinations by the Public Health service in the federal state) in order to assess the potential bias of the restriction to class level 6 for 12-year-olds, which was binding within the NOHSC since 1994. In 2016, the calibrated all 12-year-olds in the 6th grades in the federal state or about 39% of all 12-year-olds living in the state. With the inclusion of data collected by dentists not specifically calibrated in the course of the NOHSC in 2016, the total sample of 12-year-old 6th-graders increases (n = 7594, Table 1). On the level of schools, at least 82% could be reached in 2016, which means that the estimated level of confidence for a full survey of at least 70% of all schools was met [20]. Due to a comparable sample size in all the years (2008–2018), this can be assumed for all examinations. Despite the simple grouping of school types into two different categories for all federal states, caries data regarding class level in Saxony-Anhalt were presented with three different school categories for better understanding of the data (Table 3).

### 2.3. Dental Examination and Calibration of Examiners in NOHSC and Saxony-Anhalt

Caries experience in children was assessed with the DMFT-index for the permanent dentition with its single components decayed (DT), missing (MT) and filled teeth (FT) according to World Health Organization standards [16]. The dental status was entered into an available software at the community service in the state (ISGA, Octoware/easy-soft, Gudental, Micropro) or into a documentation sheet in Excel (Excel 2016). To assure confidentiality, variables like gender, age, class level, type of school, and date of examination were anonymously documented with the single components of the DMFT and initial caries (I) lesions.

All participant examiners for the NOHSC were calibrated online regarding the examination methods and the diagnosis of caries and caries experience (IDMFT). After going through the theoretical and applied test modules, the examiners were calibrated by passing a calibration module online with a randomized sequence of clinical pictures. The training and calibrations modules could be repeated as needed.

In total, 482 examiners were calibrated successfully for the NOHSC all over Germany with Kappa-values ranging from 0.65 to 1.0, and a peak of 0.85 (Figure 1).

Specifically, in the state of Saxony-Anhalt, 21 dentists were calibrated following a similar distribution compared to the national level (Figure 1). Due to routine yearly examinations of, in general, the same examiners (dentist in public health service) in Saxony-Anhalt, a similar correlation of examination quality can be assumed for the caries data for the time period 2008–2018 (Table 1).

### 2.4. Statistical Analysis and Adjustment Models

The data from the 14 public dental services in Saxony-Anhalt were assembled, processed, and analysed (descriptive and analytic) in IBM SPSS Statistics (Version 26). Caries experience (DMFT) is presented with means. A statistically significant difference was set at *p* < 0.05. For caries data, the Mann–Whitney U test was chosen to detect statistically significant differences (Table 1). Regression analyses were performed for the detection of the source and the extent of the selection bias regarding class level and school type.

In a pilot phase, a coefficient of 1.173 was found for the DMFT to adjust for the assessed selection bias due to the restriction to 6th grade in this age group of 12-year-olds. This finding was based on the caries data of Saxony-Anhalt from 2015–2016 [9]. To minimize chances for random errors based on the caries data of Saxony-Anhalt, the extrapolation was performed for the other federal states not only based on data for this one year, but with a pooled (caries data 2008–2018) coefficient of 1.142 (Table 1, Table 3 and Table 4). As different proportions of children in the NOSCH sample were examined [9], these do not fully reflect the distribution of children in 6th grade to the different school forms (Table 2). Therefore, an additional adjustment was performed based on the distribution of registered children according to type of school [15] to adjust for the assessed selection bias (Table 4). Due to a very low number of children examined in “Gymnasium” in Berlin, its adjustment in model 2 should be looked at with caution. Generally, children with missing data on caries prevalence or age were neither included in the underlying datasets of the NOHCS (a) nor the assembled data from the 14 public dental services in Saxony-Anhalt (b).

## 3. Results

Out of 96,842 12-year-olds, 76,456 attended 6th grade (78.9%), while 15,743 schoolchildren attended a class level ≤5 and 4643 schoolchildren a class level ≥7, excluding more than 21.1% of the 12-year-olds with the convenient examination strategy (Table 1). The DMFT of 12-year-olds in Saxony-Anhalt, when calculated for all pupils regardless of the grade they attend, was consistently (in all examination years) higher than the DMFT of 6th-graders only (range: +0.08 to +0.13 DMFT; equivalent to a range of +10.7% to 21.2%, Table 1). This was due to an increased DMFT of pupils attending higher (≥7) grades, but also, due to an increased DMFT in lower (≤5) grades (Table 1). Except for one year (2010), the differences in DMFT were always (in all examination years) statistically significant both in ≤5th grade as well as in ≥7th grade (Table 1). This was true while 12-year-olds in 6th grade in Saxony-Anhalt exhibited a clear decline in caries experience from 2008 to 2018 (1.0–0.52 DMFT), in line with the national trend. The mean difference in caries experience in Saxony-Anhalt between 12-year-olds in 6th grade only (n = 76.456; DMFT = 0.70, Table 1) and all 12-year-olds examined regardless of the grade attended (n = 96.842; DMFT = 0.80) was +14.2% (equivalent to +0.10 DMFT; based on pooled data 2008–2018). Most strikingly, 12-year-olds in a class level ≤ 5 and in a class level ≥ 7 exhibited 1.69 times higher and 1.64 times higher DMFT values than the NOSCH reference group, respectively. Consequently, the mean caries experience for all 12-year-olds is underestimated by 14.2% when caries data are restricted to 12-year-olds in 6th grade, as performed in the school examination of the NOHSC.

When caries data of 12-year-olds in ≤5th, 6th and ≥7th grade were separated according to the different school types attended (Table 3) an increased DMFT in ≥7th grade was found in all school types, whereas an increased DMFT in ≤5th grade was only found in “Gymnasium” and other secondary schools, but not in special needs schools. 12-year-olds in “Gymnasium” had less than half the caries experience compared to 12-year-olds in other secondary schools, and less than one third of the caries experience compared to 12-year-olds in special needs schools (0.41 vs. 0.91 vs. 1.45 DMFT; Table 3).

In a regression analysis, when “class level” and “school type” were accounted for, both factors had a statistically significant impact on DMFT, with the influence of school type being stronger (Gymnasium vs. other secondary schools + special needs schools: *p* < 0.001; −0.524 DMFT; 95% CI: −0.545 to −0.503) than the influence of attending ≥7th grade (*p* < 0.001; +0.345 DMFT; CI: 0.301–0.390) and the influence of attending ≤5th grade (*p* < 0.001; +0.279 DMFT; CI: 0.252–0.306).

Extrapolating the selection bias due to the restriction of examining only 12-year-olds in 6th grade (adjustment model 1) results in an absolute increase of 0.1 DMFT equivalent to +14.2% as compared to the crude DMFT (Table 1 and Table 4). As a result of an additional adjustment (model 2) accounting for the distribution of children to different school types in the German National Oral Health Survey in Children in 2016, the mean DMFT values of model 1 in the federal states partly decrease and partly increase with a range of −0.27 to +0.06 DMFT (equivalent to −32.4% to +12.6%). In the overall (national) sample of the 2016 NOHSC, the increase in DMFT due to grade-related adjustment (+14.2%) and the slight decrease in DMFT due to school form-related adjustment (−2.0%) add up to an overall increase in the crude DMFT of +13.6% (Table 4). Due to the small numbers of pupils examined in Gymnasium in Berlin (n = 40), school-type-related adjustments in Berlin and hence on the national level should be regarded with some caution.

## 4. Discussion

For the present analysis, caries data in 12-year-old schoolchildren attending all class levels in Saxony-Anhalt in the time period 2008–2018 were analysed in order to assess a potential selection bias within the convenient national sample restricted to 12-year-olds in 6th grade. Most importantly, the caries levels in the NOHSC are underestimated by a systematic bias due to the exclusion of a major fraction of 12-year-olds attending other class levels than 6th grade (Table 1). The underestimation of caries levels of 12-year-olds resulting from the restriction to 6th grade is mainly due to three factors, which are as follows: (a) the omission of 12-year-olds in ≤5th grade in regular schools who often are students who have repeated a previous grade and are, thus, behind their original cohort which may reflect lower socio-economic status and hence higher caries levels; (b) the omission of 12-year-olds in special needs school who often have substantially higher caries levels, but do attend another class level than grade 6; and (c) the omission of relatively few, slightly older 12-year-olds in ≥7th grade who “naturally” (in tendency older than 12 year-olds in 6th grade) have slightly higher caries levels. The additional effect of the convenient examination in different school types on the reported levels of caries experience in 12-year-olds showed varying effects in the federal state (increase or decrease in DMFT). On the national level, convenient school type selection led to a minor overestimation of DMFT by 2% (0.51 vs. 0.50 DMFT, Table 4). These mentioned aspects are discussed in the following paragraphs in detail.

The main results of this study showed that there were differences in the DMFT scores in Saxony-Anhalt when these were analysed regarding the class level as for all examination years (2008–2018), 12-year-olds in 6th grade presented significantly lower DMFT values than in all grades (range +0.08 to 0.13 DMFT; 10.7% to 21.2%; mean: +0.10 DMFT equivalent to 14.2%, Table 1). The caries levels of 12-year-olds in ≤5th grade (1.18 DMFT, Table 3) were found to be about 1.7 times higher than those of 12-year-olds in 6th grade (0.70 DMFT, Table 3). This is, at first sight, surprising since 12-year-olds in class level 5 are supposedly slightly younger than 12-year-olds in 6th grade and thus supposedly slightly less exposed to the carious process. In the German educational system, 12-year-olds usually attend 6th grade. We therefore hypothesize that among the 12-year-olds observed in ≤5th grade, there is a significant fraction of children who have, due to lower learning capacities, entered school belatedly and/or repeated one or two classes in elementary school and/or in the first two years of secondary school. This likely correlates with a lower educational background of the parents, which is known to be associated with higher caries experience and worse quality of life [21]. Within the German educational system, a strong association between social origin and children’s reading literacy exists [22], which is also reflected by higher caries experience.

The likely explanation for the higher DMFT of 12-year-olds in ≥7th grade compared to that of 12-year-olds in 6th grade, is, on the other hand, quite different. Most likely, this is due to their slightly higher age (by several months); the 12-year-olds in ≥7th grade are exposed consequently for a slightly longer time span to the carious process and thus, have a slightly higher mean DMFT, as regression analysis showed.

The two main reasons (higher DMFT in ≤5th grade due to lower learning performance and higher DMFT in ≥7th grade due to higher age) can be observed in most school types (Table 3). The specific finding in special needs schools that ≤5th-graders do not have higher caries levels can be explained by the fact that special needs schools are less competitive and grade allocation is rather arbitrary and more dependent on age than on learning performance or socio-economic background.

The caries level of children is highly dependent on the type of school attended because this, in turn, is closely associated to the socio-economic background of the children (Table 3). Our regression analysis shows that the impact of attending “Gymnasium” on the DMFT of 12-year-olds is much stronger (−0.524 DMFT) than the impact of their attending lower (+0.279 DMFT) or higher (+0.345 DMFT) grades than 6th grade.

The hypotheses on the impact of school type and class level stated in the introduction can consequently be considered true and should promote public health caries preventive efforts in a way that caries risk. Schools and children may be addressed more frequently or with more efficient measures to minimize inequalities in oral health. This is in line with the Marmot review suggesting a “proportionate universalism” for preventive actions to reduce the steepness of the social gradient in health [23]. This approach has its benefits economically as well as socially [24], as “health inequalities are not inevitable and can be significantly reduced” [23].

The comparative data for this analysis and suggestion of adjustment were obtained from the total data recorded at the national level from the recent 2016 German NOHSC, which included, among other data, the assessment of the DMFT values of 55,002 12-year-old children. The extrapolation (effect of selection bias) shows (depending on the federal state) mostly an increase in the mean DMFT of about 0.1 DMFT for all 12-year-olds (adjustment model 1; Table 4). A small but potential drawback of the study is that the DMFT levels in the NOHSC sample in Saxony-Anhalt (DMFT = 0.52) were slightly higher than the mean caries experience for all of Germany (DMFT = 0.44; [11]). This may be due to the generally lower socio-economic status of the population in this federal state, as Saxony-Anhalt was subject to major demographic changes in the wake of reunification comparable to other states of the former GDR [25]. Another possible reason for higher caries levels in Saxony-Anhalt could be the full survey evaluation instead of the sampling method used in most federal states over the course of the NOHSC [9]; e.g., a sub-analysis of the random cluster selection from the state schools’ lists compared to the full-census examinations revealed lower caries levels in the school sample for the federal state and capital Berlin for 6–7-year-olds [26] as well as for 12-year-olds (difference of −0.11 DMFT; equivalent to 14.8%) [9]. This bias likely originates in the work of the public health service addressing rather risk schools and caries risk children.

Despite the described effects of selection bias on DMFT, the reported caries data of the past and recent NOHSC should still be considered valid, as the effect sizes in the presented adjustment models are moderate and the differences in outcome do not justify switching away from the convenient examination strategy.

Socio-economic status and lower educational level are known to be relevant risk factors for higher caries levels in children in Germany [1] but also internationally [21,27]. Therefore, further adjustment according to the distribution of schoolchildren to ‘Gymnasium’ and other school types was performed (adjustment model 2; Table 4). For the DMFT in most federal states, this had a clinically rather small additional effect, despite its statistically significant impact. On the national level, the difference between the DMFT adjustment models 1 and 2 was minimal (−2%), showing that the national sample in the 2016 NOSHC was highly representative for the educational system of 6th-graders in Germany on a national level in the same year (Table 4).

The two models of adjustment taken together reveal that the crude national DMFT deriving from 6th-graders in the 2016 NOHSC underestimated the “true” national DMFT for all 12-year-olds by about 15% (increase from 0.44 to 0.50 DMFT, Table 4). This analysis shows that caries levels for 12-year-olds in the course of the NOHSC have been and still are underestimated, mainly because a relevant fraction of children with lower learning performance and higher caries levels are excluded. Although the percent of underestimation was relatively high (15%) due to presently low caries levels in 12-year-olds in Germany, this corresponded to a relatively small absolute change in DMFT (0.1).

The transferability of our results to all federal states in Germany is marginally limited as each state has their own and potentially slightly different legislations and rules for the age of entering school [28], for routine oral examinations in schools (some obligatory, some voluntary) [29], and also a variety of different types of schools [9,15]. Nonetheless, the adjusted DMFT values (adjustment model 1 and 2) result in an identical finding for caries experience in 12-year-olds, similar to another representative national study from Germany which applied a different methodological sampling approach (0.50 DMFT vs. 0.5 DMFT [1]). Consequently, the two adjustment methods proposed in this study (accounting for grade-related and school type-related selection biases) should also be taken into consideration for previous NOHSCs (1994–2009 [13]) and in future examinations which may likely still be performed due to the costs, comparability, and feasibility of applying the convenient examination strategy.

Irrespective of the assessment of the data quality, the NOHSC being embedded in the regular school dental screening programme for oral health likely has some impact on oral health itself. Such regular and widespread screenings via local public dentists aim to identify potential dental problems like caries before symptomatic disease presentation to initiate prevention for risk groups and/or therapeutic dental care for affected children [30]. These strategies to promote oral health are important, especially for children who are difficult to reach outside of the school in contrast to children and adolescents who regularly attend dental visits including individual caries prophylaxis anyway [31]. Moreso, valid caries data are important for the assessment of the efficacy of integrating oral health preventive measures [32,33].

These results (especially DMFT) assessed on the basis of very large sample sizes (n > 55,000) show that the methods of the NOHSC are highly robust but have been and still are subject to a systematic error as presented in this study. Other international and national studies on caries prevalence in children struggle with the risk of sampling/selection bias as well [1,12]. Nonetheless, the achievement of representativeness of the sample for the population always stays a major goal and this was the motivation for carrying out this study to assess the impact of selection bias. However, avoiding this systematic bias within future NOHSC would lead to an unjustified increase in time, costs, and efforts and, therefore, should rather be statistically adjusted with the presented coefficient. To our knowledge, despite there are some other (serial) cross-sectional studies on caries prevalence in adolescents in Europe alike the NOHSC [34,35], no other epidemiological study has investigated specific effects of potential bias nor has performed adjustments of the reported caries levels.

Consequently, other national caries studies worldwide using the robust marker of DMFT in low caries prevalence populations should also rather statistically adjust for systematic selection bias related to socio-economic background than increasing efforts in examination strategy.

## 5. Conclusions

In conclusion, restricting dental examination in 12-year-olds to 6th-graders excludes more than 20% of 12-year-olds and underestimates caries levels within the NOHSC by about 15%. Furthermore, selection bias regarding school types was shown to under-/overestimate certain regional caries levels. Consequently, mean DMFT values in the 2016 NOHSC should be adjusted from 0.44 to 0.50 DMFT for all 12-year-olds in Germany. This proposed two-step adjustment of crude regional DMFT should be transferred also for past and future NOHSC examinations. Despite the statistically significant impact of selection bias in the NOHSC on caries experience, the overall interpretation of the epidemiologic situation regarding caries in children is only minimally affected.

## Figures and Tables

**Figure 1 ijerph-21-00467-f001:**
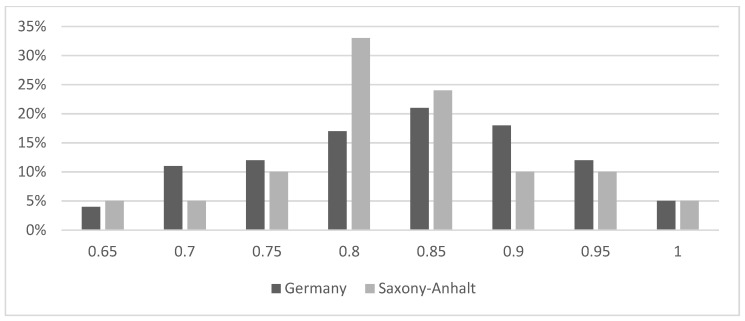
Distribution of Kappa values of all the participating examiners in the German National Oral Health Survey 2016 (n = 482) as compared to the calibrated examiners in Saxony-Anhalt (n = 21).

**Table 1 ijerph-21-00467-t001:** Caries experience (DMFT) in 12-year-olds in Saxony-Anhalt in the time period 2008–2018 in relation to the attended class level.

Year of Examination	≤5th Grade ^1^			6th Grade ^2^		≥7th Grade ^2^			All Grades ^3^		Difference of DMFT in All Grades Minus DMFT in 6th Grade	
	Examined Children	DMFT		Examined Children	DMFT	Examined Children	DMFT		Examined Children	DMFT		
	*N*	Mean	*p*-Val. ^4^	*N*	Mean	*N*	Mean	*p*-Val. ^4^	*N*	Mean	Absolute	In Percent
**2008**	1019	1.78	<0.001	5370	1.00	522	1.26	0.002	6911	1.14	0.13	13.4%
**2009**	1322	1.57	<0.001	6640	0.89	499	1.33	<0.001	8461	1.02	0.13	14.7%
**2010**	1102	1.52	<0.001	6553	0.88	638	1.00	0.135	8293	0.97	0.09	10.7%
**2011**	1256	1.35	<0.001	6925	0.80	625	0.94	0.039	8806	0.89	0.09	11.0%
**2012**	1343	1.22	<0.001	8082	0.71	650	1.01	<0.001	10,075	0.80	0.09	12.3%
**2013**	1544	1.16	<0.001	7148	0.71	563	1.10	<0.001	9255	0.81	0.10	13.9%
**2014**	1621	0.99	<0.001	7452	0.59	275	1.57	<0.001	9348	0.69	0.10	16.4%
**2015**	1595	0.92	<0.001	7158	0.60	269	1.28	<0.001	9022	0.67	0.08	12.9%
**2016**	1593	1.02	<0.001	7594 *	0.54	240	1.22	<0.001	9427	0.63	0.10	18.4%
**2017**	1667	0.93	<0.001	7227	0.55	194	1.22	<0.001	9088	0.64	0.08	14.9%
**2018**	1681	0.96	<0.001	6307	0.52	168	1.39	<0.001	8156	0.63	0.11	21.2%
**2008–2018**	15,743	1.18		76,456	0.70	4643	1.15		96,842	0.80	0.10	14.2%

^1^ In elementary schools and special needs schools. ^2^ Reference group of 12-year-olds in the NOHSC; in Gymnasium, other secondary schools (regular secondary schools, lower secondary schools, integrated comprehensive schools, school types with several courses of education, orientation levels independent of school type, independent Waldorf schools), and special needs schools). ^3^ In all school forms. ^4^ Testing of statistically significant difference in DMFT compared to DMFT in reference group (6th grade) was performed with Mann-Whitney-U-Test. * The number of children examined in 6th grade and the DMFT shown in Table 1 for the year 2016 differ slightly from those published for Saxony-Anhalt in the 2016 NOHSC [9]. This is due to the fact that data from a small fraction of dentists in Saxony-Anhalt who were not specifically calibrated just prior to the NOHSC are included in Table 1, whereas they were excluded in the 2016 NOHSC, even if they had been calibrated in earlier years.

**Table 2 ijerph-21-00467-t002:** Registered 6th-graders ^1^ and examined 12-year-old 6th-graders ^2^ differentiated by school type category in the different German federal states in the 2015–2016 school year.

Federal State	Gymnasium ^3^				Other Schools ^4^			
	Registered 6th-Graders School Year 2015/2016 ^1^		Examined 12-Year-Olds in 6th Grade in NOHSC 2016 ^2^		Registered 6th-Graders School Year 2015/2016 ^1^		Examined 12-Year-Olds in 6th Grade in NOHSC 2016 ^2^	
	n	%	n	%	n	%	n	%
**BW**	40,403	38.8	612	39.9	63,639	61.2	922	60.1
**BY**	41,767	36.3	310	25.3	73,187	63.7	917	74.7
**BE**	2293	7.2	40	0.6	29,701	92.8	6411	99.4
**BB**	869	4.4	254	3.7	19,090	95.6	6665	96.3
**HB**	1412	27.0	315	24.5	3810	73.0	972	75.5
**HH**	7669	50.8	1731	52.4	7432	49.2	1574	47.6
**HE**	24,383	44.7	502	26.0	30,118	55.3	1432	74.0
**MV**	697	5.3	307	16.5	12,359	94.7	1557	83.5
**NS**	31,734	41.1	797	53.9	45,519	58.9	682	46.1
**NRW**	65,247	38.9	2348	43.4	102,461	61.1	3062	56.6
**RP**	14,930	41.4	1292	46.0	21,127	58.6	1517	54.0
**SL**	3190	40.5	756	42.1	4684	59.5	1039	57.9
**SN**	13,007	40.0	919	41.0	19,490	60.0	1325	59.0
**ST**	7357	44.1	2773	43.4	9309	55.9	3618	56.6
**SH**	9975	38.2	1013	36.5	16,159	61.8	1764	63.5
**TH**	6904	40.4	3052	40.5	10,194	59.6	4487	59.5
**Germany**	271,837	36.7	17,021	31.0	468,279	63.3	37,944	69.0

BW: Baden-Wuerttemberg; BY: Bavaria; BE: Berlin; BB: Brandenburg; HB: Bremen; HH: Hamburg; HE: Hessen; MV: Mecklenburg-Western Pomerania; NS: Lower Saxony; NRW: North Rhine-Westphalia; RP: Rhineland-Palatinate; SL: Saarland; SN: Saxony; ST: Saxony-Anhalt; SH: Schleswig-Holstein; TH: Thuringia. ^1^ Without differentiation by annual age groups; data source: Statistik der allgemeinbildenden Schulen [15]. ^2^ Only 12-year-olds; source of data: NOHSC 2016 [9]. ^3^ Gymnasium is attended mainly by schoolchildren of higher learning performance and more privileged social background; Gymnasium G8, Gymnasium G9. ^4^ Secondary schools, lower secondary schools, integrated comprehensive schools, school types with several courses of education, orientation levels independent of school type, independent Waldorf schools, special needs schools.

**Table 3 ijerph-21-00467-t003:** Mean DMFT and its single components (pooled data 2008–2018) in 12-year-olds in Saxony-Anhalt depending on the attended school type and class level in Saxony-Anhalt for comparison to the NOHSC group (12-year-olds in 6th grade only).

Saxony-Anhalt		Examined 12-Year-Olds	DMFT	D	M	F
School Type and Class Level		n	Mean	Mean	Mean	Mean
**Gymnasium**	≤5th grade	712	0.74	0.14	0.03	0.56
	**6th grade**	**32,416**	**0.40**	**0.06**	**0.01**	**0.33**
	≥7th grade	1004	0.48	0.06	0.02	0.40
	Total	34,132	0.41	0.06	0.01	0.34
**Other schools** ^1^	≤5th grade	11,254	1.16	0.39	0.07	0.71
	**6th grade**	**38,691**	**0.84**	**0.21**	**0.04**	**0.58**
	≥7th grade	1145	0.86	0.13	0.04	0.68
	Total	51,090	0.91	0.25	0.05	0.61
**Special needs schools**	≤5th grade	3777	1.30	0.43	0.10	0.77
	**6th grade**	**5349**	**1.50**	**0.48**	**0.11**	**0.91**
	≥7th grade	2494	1.56	0.51	0.12	0.92
	Total	11,620	1.45	0.47	0.11	0.87
**All school forms**	≤5th grade	15,743	1.18	0.38	0.07	0.72
	**6th grade**	**76,456**	**0.70**	**0.16**	**0.03**	**0.50**
	≥7th grade	4643	1.15	0.32	0.08	0.75
	Total	96,842	0.80	0.21	0.04	0.55
Difference of mean value in all classes to 6th grade in %			**14.2%**	**26.1%**	**25.0%**	**9.4%**

^1^ Predominantly secondary schools, comprehensive schools, community schools, but also Waldorf schools and elementary schools. Reference data for 6th-graders as used in NOHSC is highlighted in bold.

**Table 4 ijerph-21-00467-t004:** Published mean caries experience in the NOHSC sample of 12-year-olds in 6th grade in all federal states in Germany (2015–2016) and their adjusted caries values for 12-year-olds based on caries data of Saxony-Anhalt and the distribution of children to the different school forms.

German Federal State	DMFT in 12-Year-Olds in 6th Grade (Reported in NOHCS)						Adjustment Models for DMFT		
	Gymnasium		Other Secondary Schools ^1^		All School Forms		Model 1	Model 2	
	n	DMFT	n	DMFT	n	“Crude” DMFT	Adj. DMFT with Coefficient of 1.142 for 12-Year-Olds in All Class Levels ^2^	Adjusted DMFT after *Additional* Correction for School Forms ^3^	Deviation of DMFT in % from “Crude” DMFT after Adjustments 1 and 2
**BW**	612	0.16	922	0.53	1534	**0.38**	0.44	**0.45**	17.7
**BY**	310	0.40	917	0.69	1227	**0.62**	0.70	**0.67**	8.1
**BE**	40	0.08	6411	0.75	6451	**0.74**	0.85	**0.57 ^4^**	−22.8 ^4^
**BB**	254	0.19	6665	0.49	6919	**0.48**	0.55	**0.44**	−9.6
**HB**	315	0.30	972	0.76	1287	**0.65**	0.74	**0.68**	4.2
**HH**	1731	0.29	1574	0.49	3305	**0.39**	0.44	**0.48**	23.6
**HE**	502	0.21	1432	0.44	1934	**0.38**	0.43	**0.40**	6.8
**MV**	307	0.40	1557	0.47	1864	**0.46**	0.52	**0.51**	10.8
**NS**	797	0.29	682	0.61	1479	**0.44**	0.50	**0.56**	28.6
**NR**	1491	0.24	1805	0.51	3296	**0.39**	0.44	**0.47**	20.9
**RP**	1292	0.16	1517	0.30	2809	**0.24**	0.27	**0.28**	20.5
**SL**	756	0.19	1039	0.33	1795	**0.27**	0.31	**0.32**	17.4
**SN**	919	0.26	1325	0.57	2244	**0.44**	0.50	**0.52**	17.6
**ST**	2773	0.29	3618	0.69	6391	**0.51**	0.59	**0.62**	20.1
**SH**	1013	0.18	1764	0.42	2777	**0.34**	0.38	**0.38**	14.0
**TH**	3052	0.25	4487	0.57	7539	**0.44**	0.50	**0.51**	17.3
**WL**	857	0.27	1257	0.49	2114	**0.40**	0.46	**0.46**	16.6
**GER** ^5^	**17,021**	**0.25**	**37,944**	**0.55**	**54,965**	**0.44**	0.51	**0.50**	**13.6**

^1^ Secondary schools, lower secondary schools, integrated comprehensive schools, school types with several courses of education, orientation levels independent of school type, independent Waldorf schools, special needs schools. ^2^ According to the presented large study population in Saxony-Anhalt (pooled data 2008–2018), the DMFT value of 12-year-olds is 1.142 times higher when all class levels are included than when only the 6th-graders are considered (Table 1). ^3^ The crude DMFT values in the 2 school form categories in each federal state were weighted with the actual frequency of schoolchildren attending these school categories in Germany. The actual frequency was calculated on the basis of the number of pupils in 6th grades in Germany in the 2015–2016 school year. The fraction of 6th-graders in the two school form categories in the 2015–2016 school year were 36.7% (Gymnasium): 63.3% (other general education schools) (Table 2 and Table 4). This was then multiplied by the factor 1.142 from adjustment 1, resulting in the combined adjustment 2. ^4^ The DMFT value after adjustment 2 in Berlin is highly uncertain due to a very low sample size for school category “Gymnasium”. Therefore, exceptionally for Berlin, the DMFT of model 1 (0.84 DMFT) was used for calculation of the mean adjusted (model 2) caries experience in Germany. ^5^ GER (Germany): The DMFT values for Germany were calculated as follows: (a) For DMFT in Gymnasium and secondary schools, and all school forms the mean values were weighted with the actual numbers of 6th-graders in the federal states in the school year 2015/2016 (Table 2); (b) DMFT in Adj. 1: Multiplication of the German value of the DMFT for all schools determined by the factor of adjustment 1 (1.142, cf. Table 1); (c) DMFT in Adj. 2: Sum of the crude values for Germany at Gymnasiums and other secondary educational schools were weighted with the actual (national) frequencies of 6th-graders in Gymnasiums and other secondary educational schools (cf. Table 2) multiplied by the factor of adjustment 1: (0.25 × 0.3673 + 0.55 × 0.6327) × 1.142. (d) All values for Germany were rounded to 2 decimal places. Bold number highlight “crude” DMFT and “adjusted” DMFT to facilitate readability and comparison of absolute mean caries values.

## Data Availability

Data is available in publicly accessible repositories that do not issue DOIs. Detailed data regarding the German National Oral Health Survey in Children (NOHSC) from the 2015–2016 school year for all federal states in Germany [9,10] is available within the national report online for free https://daj.de/wp-content/uploads/2024/02/Epi_final_BB0103_final_Druckvorbereitung.pdf (accessed on 4 April 2024). Assembled data from the 14 public dental services in Saxony-Anhalt from the annual oral examinations accompanying group prophylaxis in schools from the years 2008–2018 for all 12-year-olds irrespective of the class level was collected by the Landesamt für Verbraucherschutz Sachsen-Anhalt (LAV) and is presented in this article in detail (Table 1 and Table 3). Their use in the present study was kindly authorized by the Landesarbeitsgemeinschaft für Jugendzahnpflege (LAJ) Sachsen-Anhalt e. V. Data from the German federal statistical office (https://www.destatis.de/DE/Home/_inhalt.html, accessed on 4 April 2024) on distribution of schoolchildren in 6th grade to the different types of schools in each federal state is publicly available on request (https://www-genesis.destatis.de/genesis/online?operation=table&code=21111-0005&bypass=true&levelindex=1&levelid=1598437198298#abreadcrumb, accessed on 4 April 2024) [15,19].

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
