# Peer review of "Influence of School Type and Class Level on Mean Caries Experience in 12-Year-Olds in Serial Cross-Sectional National Oral Health Survey in Germany—Proposal to Adjust for Selection Bias"

_ijerph, 2024, doi:10.3390/ijerph21040467_

Round 1
Reviewer 1 Report
Comments and Suggestions for Authors
Firstly, the abstract is so long, it must be resume, the material and methos and the results are son long.
Related to the end of the introduction, rewrite the aim, it is confuse.
In relation to material and methods, in general needs to be explained better, because It´s very confuse, the methodology used is not clear. In addition, from line 132 to 139 it must be resumed.
In my opinion, the justification and backgrownd in the material and methods section it should be in introduction section, not there.
There isn´t inclusion and exclusion criteria, and I think is an important part of a paper. There is no sample size calculation eather.
I have some doubts: Socioeconomic level and parental profession has been taken into account? Has been consider the possibility that the children could change the school? Have been carried out surveys to parents?
The explanation of the table 1 must be resumed of the materials and methods. Please It is recommended to add a part of the explanation at introduction section.
In my opinion it is necessary to create an appendix section. In this part, It should be add tables 1 and 3 and figure 1.
In line 193, it is mentioned that “the estimated level of confidence for a full survey of at least 70%”. In my opinion, It is a low percentage. I don't know if it could be modified.
From line 437 to 442, it is a personal opinion, It ´s better to add un discussion not in conclusion
Author Response
Dear Reviewer,
please, find below our responses to you comments.
Firstly, the abstract is so long, it must be resumed, the material and methos and the results are so long.
Yes, you are right, we shortened the abstract especially in the parts material and methods and the results.
Related to the end of the introduction, rewrite the aim, it is confuse.
We rephrased the aim of the study to be less confusing, but (hopefully) still precise.
In relation to material and methods, in general needs to be explained better, because It´s very confuse, the methodology used is not clear. In addition, from line 132 to 139 it must be resumed.
We rephrased the aim of the study to be less confusing but still precise.
In my opinion, the justification and background in the material and methods section it should be in introduction section, not there.
We understand the concern. We discussed this again within the author group and we decided not to move this part from the material and methods section to the introduction section, as it is in our eyes part of the methodology and important to describe briefly the approach used for data collection of the underlying data (see also comment to next remark).
There isn´t inclusion and exclusion criteria, and I think is an important part of a paper. There is no sample size calculation eather.
Yes, we added now the inclusion criteria for NOHSC to the methodology, which underlines that this part should be presented in the methods section. We didn’t mention a specific sample size calculation as the samples were thought be high for all federal stats. In the end it was more a matter of representativity of the examined sample, which we assess with this adjustment study presented in this manuscript.
I have some doubts: Socioeconomic level and parental profession has been taken into account? Has been consider the possibility that the children could change the school? Have been carried out surveys to parents?
Socioeconomic aspects of the children are covered in this manucsipt and as well as in the underlying examination (e.g. NOHSC) solely by correlation of DMFT and the attended school or attended class level. No additional surveys of the parents have been carried out as the data are collected from routine examination data (in schools) of the dental public health sector.
Yes, you are right, children may change the school, but this is in our eyes not relevant as the caries data is obtained from (serial) cross-sectional examinations and not from longitudinal data. Moreso, we have an enormous sample, so that this potential bias would be minimal.
The explanation of the table 1 must be resumed of the materials and methods. Please It is recommended to add a part of the explanation at introduction section.
We disagree to resume the explanation of table 1 as this table forms one major basis for the proposed later adjustments. Still, we have adopted your recommendation in the way that we have added a small explanation in the introduction of the manuscript.
In my opinion it is necessary to create an appendix section. In this part, It should be add tables 1 and 3 and figure 1.
We understand your recommendation, but we would like to refrain from it, as we think – in line with the other 3 reviewers - that they shall be part of the main manuscript.
In line 193, it is mentioned that “the estimated level of confidence for a full survey of at least 70%”. In my opinion, It is a low percentage. I don't know if it could be modified.
Yes, this is what we have written according to the mentioned cited paper. 70% of a full survey might sound not so much, but considering the enormous sample sizes of several ten-thousands of examined children used as the underlying date for adjustments it is statistically sufficient. We want to point out here again, that this is also one main apsect we want prove with this adjustment paper: On a national level the used methods for examination give a highly sound impression of the caries data and only minor changes can be seen on the level of federal states.
From line 437 to 442, it is a personal opinion, It ´s better to add un discussion not in conclusion
Yes, we have moved this part as you suggested from conclusion to discussion section.
Thank you very much for your recommendations and the time you invested for careful reading of the paper!
Reviewer 2 Report
Comments and Suggestions for Authors
This is a paper of importance to the field of epidemiology, The study design is an ecological study. The entire study methodology is difficult to understand as it followed no logical order.
Please can the methodology be re-written using the STROBE guideline as a guide. Please submit the revised manuscript along with the STROBE guidelines. When there is clarity with the methodology, then the results, discussion and conclusion can be read meaningfully
Comments on the Quality of English LanguageGood enough
Author Response
Dear Reviewer,
please, find below our responses to you comments.
This is a paper of importance to the field of epidemiology, The study design is an ecological study.
Thank you very much!
The entire study methodology is difficult to understand as it followed no logical order.
We disagree regarding the logical order, but we understand that due to the complexity of the methodology of this “Statistical / epidemiological paper” it is difficult to understand, so we put efforts into increasing the readability and understandability of the section (see main manuscript for the adjustments made).
Please can the methodology be re-written using the STROBE guideline as a guide. Please submit the revised manuscript along with the STROBE guidelines. When there is clarity with the methodology, then the results, discussion and conclusion can be read meaningfully
We understand your recommendation. We are well aware of STROBE: Strengthening the reporting of observational studies in epidemiology. Nonetheless, the fact that we used data collected from observational studies in epidemiology, this paper itself did not have the aim to report again the available data but to present an adjustment strategy and to evaluate the magnitude of the selection bias in this studies. Nonetheless, we made some changes in the methods sections, so that this aspect is now easier to find and to understand.
Comments on the Quality of English Language
Good enough
Thank you!
Reviewer 3 Report
Comments and Suggestions for Authors
No background on your abstract.
The introduction takes no other research as the basis of your manuscript.
Tables do not have significant values (p values)
In the discussion section, Author must add some research as a comparison material
Add more recent references in the last 10 years
Author Response
Dear Reviewer,
please, find below our responses to your comments.
No background on your abstract.
We have removed the background from the abstract and shortened the abstract on top of that according to another reviewers comments.
The introduction takes no other research as the basis of your manuscript.
Yes, this is mainly the point of the paper, but we have added some aspects to highlight also the international relevance of the paper, that it may be advisable to priorize adjustments for systematic selection bias associated with socio-economic backgrounds over fine-tuning of examination strategies.
Tables do not have significant values (p values)
This is actually not correct for all tables.. E.g. in Table 1 we have the columns of “Sign.” referring to it. This was just labelled differently. We changed it now to p-value.
In the discussion section, Author must add some research as a comparison material
To our knowledge, a comparable research has never been performed as the data basis and the adjustment strategy of this paper are unique!
Add more recent references in the last 10 years
We added another 2 recent references, but we are happy if you could provide us with any if you have a specific one that fits well.
Thank you very much for your recommendations and the time you invested for careful reading of the paper!
Reviewer 4 Report
Comments and Suggestions for Authors
Dear Authors,
Your manuscript is highly intriguing and significant, particularly in nations with a higher DMF Index. It would be advisable to prioritize adjustments for systematic selection bias associated with socio-economic backgrounds over fine-tuning examination strategies.
Good Luck
Author Response
Dear Authors,
Your manuscript is highly intriguing and significant, particularly in nations with a higher DMF Index. It would be advisable to prioritize adjustments for systematic selection bias associated with socio-economic backgrounds over fine-tuning examination strategies.
Good Luck
Thank you very much for your recommendations and the time you invested for careful reading of the paper! As we liked your comment very much on “priorizing adjustments for systematic selection bias associated with socio-economic backgrounds over fine-tuning examination strategies”, we felt obliged to add this well-phrased aspect to our paper!
Round 2
Reviewer 1 Report
Comments and Suggestions for Authors
Dear Authors
After reviews the article has improved.
I would only add some comparison if it exists with other similar studies in other European countries, which there is nothing about it.
Author Response
After reviews the article has improved.
I would only add some comparison if it exists with other similar studies in other European countries, which there is nothing about it.
After reviews the article has improved.
I would only add some comparison if it exists with other similar studies in other European countries, which there is nothing about it.
Thank you.
We added two more references (see below) on studies similar to the NOHSC in Europe for comparison.
- Grieshaber, A.; Waltimo, T.; Haschemi, A.A.; Bornstein, M.M.; Kulik, E.M. Dental caries and associated factors in 7-, 12- and 15-year-old schoolchildren in the canton of Basel-Landschaft, Switzerland: Changes in caries experience from 1992 to 2021. Int. J. Paediatr. Dent. 2024, 34, 169–178, doi:10.1111/ipd.13122.
- Schuller, A.A.; Vermaire, J.H.; Verrips, G.H.W. Kies-voor-Tandenonderzoek 2017: cariëservaring bij 11-jarigen in Nederland. Ned. Tijdschr. Tandheelkd. 2020, 127, 42–50, doi:10.5177/ntvt.2020.01.19076
Reviewer 2 Report
Comments and Suggestions for Authors
Please submit the revised manuscript along with the STROBE guidelines.
Comments on the Quality of English LanguageGood enough
Author Response
Please submit the revised manuscript along with the STROBE guidelines.
Thank you very much for yor time to review our paper.
We went through the Strobe checklist again and added minor aspects according to the STROBE guidelines in the main manuscript. Moreso, we submit the STROBE checklist as PDF.
